# Effects of supernova induced soft X-rays on middle and upper atmospheric nitric oxide and stratospheric ozone

[1]David E. Siskind, [2]McArthur Jones Jr., [2,3]Jeffrey W. Reep

[1]Computational Physics Inc., Springfield, VA, USA

[2]Space Science Division, Naval Research Laboratory, Washington, DC, USA

[3] now at Institute for Astronomy, University of Hawaii at Mānoa, Pukalani, HI, USA

Corresponding authors:

David Siskind (dsiskind@cpi.com)

McArthur Jones Jr. (mcarthur.jones16.civ@us.navy.mil)

## Abstract

We provide a quantitative test of the recent suggestion (Brunton et al., 2023) that supernovae could significantly disrupt ozone layers of Earth-like planets through a multi-month flux of soft X-rays that produce ozone-destroying odd nitrogen (e.g. NO and $NO_2$). Since soft X-rays do not directly penetrate down to the ozone layer, this effect would be indirect and require downward transport of NOx from the mesosphere. Mirroring previous studies of the indirect effects of energetic particle precipitation (EPP-IE), we call this the X-ray Indirect Effect (Xray-IE). We use the NCAR Thermosphere-Ionosphere-Mesosphere-Electrodynamics General Circulation Model (TIME-GCM) to simulate the production of NO and its transport into the stratosphere. We model the soft X-ray flux as if it were a multi-month long solar flare and use our previously developed solar flare model to simulate the soft X-ray enhancement. Our results yield significant enhancement in stratospheric odd nitrogen, most dramatically in the Southern Hemisphere. The strongest global effects are seen in the upper stratosphere at pressure surfaces between 1-3 hPa (about 42-48 km) consistent with previous observations of the EPP-IE. We then use a detailed stratospheric photochemistry model to quantify the effects of this NOx enhancement on ozone. Widespread ozone reductions of 8-15% are indicated; however, because these are limited to the upper edges of the ozone layer, the effects on the ozone column are limited to 1-2%. We thus conclude that the effects of a multi-month X-ray event on biologically damaging UV radiation at the surface is also likely to be small.

## 1. Introduction

As discussed by Airapetian et al., (2019) and summarized by Garcia-Sage (2023), the explosion of new discoveries of exoplanets and the search for life in the universe as led to increased recent interest in how space weather can influence the climate and habitability of the earth and possible life-bearing exoplanets. As the above articles discuss (see also Kahler and Ling, 2023), these extreme space weather events can include solar/stellar flares, coronal mass ejections, solar/stellar energetic particles (SEPs) and/or cosmic rays. There is, however, a parallel line of inquiry that has long considered the effects of supernovae on planetary biospheres (Gehrels., et al., 2003). As we will discuss, there is significant conceptual overlap in the specific mechanisms, which is a motivation for our present study.

Recently Brunton et al. (2023) have proposed a new mechanism by which supernovae could threaten the existence of planetary biospheres. The classical mechanisms have traditionally invoked ozone depletion either due to gamma ray emission which would occur promptly (within 100 days) with the event, or from cosmic ray fluxes which could be emitted over a period on the order of 10-100 years (Gehrels, et al., 2003). Brunton et al., (2023) suggest a third mechanism from enhanced X-ray emissions that might result from interactions between the supernova blast wave and the local interstellar medium. They present observed light curves showing X-ray emissions occurring over periods ranging from 6 months to several years after the initial eruption. They suggest that these emissions might represent a heretofore unexplored mechanism for planetary ozone destruction.

An important consideration for understanding the effect of enhanced X-rays on the ozone layer, which Brunton et al (2023) discuss, is the fact that X-rays with energies less than 10-20 keV are absorbed in the mesosphere, above the ozone layer. While Brunton et al., recognize that there may be X-ray emission from a supernova with greater energies, much of their data is limited to these softer X-rays. As a result, they suggest that the effect of X-rays would be more indirect and they quote some aeronomic studies (Solomon et al., 1982; Randall et al., 2006) of how perturbations to nitric oxide in the mesosphere and lower thermosphere could be transported down to the middle atmosphere where they would catalytically lead to ozone loss. Conventionally this coupling mechanism is due the production of nitric oxide (NO) in the auroral zones near 100 km altitude by energetic electron impact on $N_2$ followed by descent through the mesosphere into the stratosphere under the cover of polar night, which limits the dissociation of the enhanced NO by UV sunlight. Randall et al., (2007) labeled this as the Energetic Particle Precipitation Indirect Effect (EPP-IE). Here, motivated by Brunton et al.'s hypothesis, we consider an analogous indirect effect on stratospheric odd nitrogen and ozone from continual soft X-ray influx, which we dub the "X-ray IE".

Brunton et al. (2023) provide estimates for the total amount of X-ray energy that might threaten planetary ozone layers and compared them to the integrated energy emitted by a multi-year solar flare. Specifically, they argue that a so-called Carrington flare (X45, i.e., $4.5 \times 10^{-3}$ W m$^{-2}$), near the upper limit of flare energy release by the Sun (see e.g., Cliver et al 2022), would have to persist for 2.8 years to provide the requisite energy. Using this analogy, we will use an existing solar flare model (Siskind et al., 2022) and consider the consequences of previously considered solar flares extending for over a year. We will show how the X-ray IE can lead to a significant influx of nitric oxide entering the stratosphere and quantitatively model to what extent this influx could reduce ozone abundances. Ultimately, we conclude that due to the specifics of how NO is transported in the middle atmosphere, while significant effects are probable, the global destruction of the Earth's ozone layer is less likely.

The general outline of the paper is as follows. In Section 2, we introduce the solar flares that form the basis of our study, look at the initial response of lower thermospheric NO and compare our calculations with previously published observations of the nitric oxide response to solar flare. In Section 3 we document the descent of this flare-produced NO down through the mesosphere using a three-dimensional model of chemistry and transport of the middle and upper atmosphere

(the NCAR Thermosphere Ionosphere Mesosphere Electrodynamics General Circulation Model
(TIME-GCM)). To validate the X-ray IE we will put it into context of our calculated EPP-IE
which can be compared with the extensive literature on that topic. Finally, in Section 4, we
perform photochemical modeling of the sensitivity of stratospheric ozone to the various
enhancements in middle atmospheric nitric oxide suggested by the TIME-GCM. One limitation
that we will discuss is that the 30 km bottom boundary of the TIME-GCM is right at the peak of
the ozone layer. Thus, ~~our~~ photochemical simulations are required to be able to extrapolate down
to encompass the entire ozone column.

**2. Solar Flare and thermospheric NO modeling**
2.1. Solar Flare modeling
Our approach follows the suggestion of Brunton et al. (2023), namely, to model the multi-month
soft X-ray flux as if it were a solar flare that lasted for months rather than the 30-60 minutes
which is typical (cf. Rodgers et al., 2010; Table 3; also Reep et al., 2023). The advantage of this
approach is that it allows us to utilize existing flare spectra (Siskind et al., 2022). These spectra
were developed with the NRLFLARE model, a physical model of solar flare irradiance, which
uses a series of flaring loop simulations to reconstruct the soft X-ray light curves of both
GOES/XRS channels, and from those loop simulations, synthesizes full spectra from
approximately 0.01 to 200 nm (Reep et al 2020; Reep et al 2022).  The ratio of the two
GOES/XRS channels is commonly used as a proxy for temperature, which the model uses to
derive heating rates to drive those simulations (see e.g. Garcia 1994).  The loop simulations are
run with the open-source radiative hydrodynamics code HYDRAD (Bradshaw & Cargill 2013;
Reep et al 2019, https://github.com/rice-solar-physics/HYDRAD), which solves the Navier-
Stokes equations for plasma constrained to travel along a magnetic flux tube.  The full model and
spectral synthesis are described in detail in Reep et al 2022.
NRLFLARE was designed to reproduce X-ray spectra from solar flares, so it is important to
discuss the differences and similarities to supernova X-ray spectra.  In both cases, the spectra in
soft X-rays (around 1 to 20 keV or so) are dominated by optically thin thermal bremsstrahlung
emission with a power law shape, with notable line emissions from hot ions such as Fe XXV (a
prominent line at 6.7 keV appears in spectra of both).  There are two important differences.
First, the elemental abundances are not the same, which will cause the relative strength of the
emission (particularly line emission) to differ.  Second, solar flares are expected to be in
collisional equilibrium, while supernova remnants have low enough densities that the collisional
timescale is long, so they are typically not in equilibrium.  See the reviews by Vink (2012) for X-
ray emissions in supernovae and Fletcher et al. (2011) for solar flares (Sections 6 of both
reviews).  For the purposes of calculating NO production, the exact spectral shape is less
important than the total soft X-ray energy input driving the atmospheric response. A key
assumption is that we are essentially ignoring wavelengths less than 0.05 nm. As discussed by
Brunton et al. (2023) these wavelengths would be absorbed much more directly into the
stratospheric ozone layer. Older studies (cf. Ejzak et al., 2007) did include these wavelengths and
this inclusion, as noted by Brunton "complicates any direct extrapolation" of those results when
considering a purely soft X-ray event, as we do here. Our work is the first to use a model of the
stratosphere, mesosphere and thermosphere to explicitly consider how the indirect effects of
enhanced soft X-rays could affect global ozone.
One of the main subjects of the Siskind et al., (2022) paper was the September 10, 2017 X8.3
flare and a spectrum at flare peak was presented in that paper. We will use that as our primary
case. Table 1 summarizes key aspects of that flare that are relevant for this paper. First, it is
important to note that in 2020, NOAA removed a 0.7 recalibration that had historically been
applied to GOES 13-15 data (cf.
https://ngdc.noaa.gov/stp/satellite/goes/doc/GOES_XRS_readme.pdf; also Reep et al., 2022)
Thus, the true X-ray irradiance for older flares is 1/0.7 brighter. This means that the 2017 flare,
originally labeled as 8.3 in Siskind et al. 2022 and earlier works (Qian et al., 2019; Redman et
al., 2018) should be re-classified as X11.8. Table 1 shows the calculated peak energy by the
NRLFLARE model as being about 12% greater than observed by GOES, thus effectively making
this flare an X13.3 event. We will thus use the label "X13" to describe this event as we discuss
our atmospheric simulations.
**Table 1**

| Event | NOAA class | Calculated 0.1-0.8 X-class with NRLFLARE | Calculated energy flux, 0.1-1.0 nm (W/m$^2$) | Calculated energy flux, 1-2 nm (W/m$^2$) | Integrated energy $\geq$ 1 keV after 1 year (kJ/m$^2$) |
|---|---|---|---|---|---|
| Sept 10, 2017 | 11.8 | 13.3 | 1.55e-3 | .0017 | 64.4 |
| Oct 28, 2003 | X25 | X27 | .004 | .007 | 171.4 |


Table 1 also shows the integrated energy in several energy bins. The division into 0.1-1.0 nm and
1-2 nm bins is to compare with the calculations of Rodgers et al., (2010), discussed below. The
final column extrapolates our flare duration to a year. In particular, it shows that if we assumed
the X13 flare persisted for an entire year, it would deliver 64.4 kJ/m$^2$ to the atmosphere. This is
less than the 400 kJ/m2 that Brunton et al., (2023) use as a critical threshold for ecologically
destructive X-ray energy input. We will therefore also consider the energy input from a spectrum
calculated for the October 28, 2003, the so-called Halloween event. The effects of this flare on
thermospheric nitric oxide were first discussed by Rodgers et al., 2010 and we will compare our
calculations to theirs. Again, due to the NOAA recalibration, this flare, which was originally
classified as X18, should really be classified as X25. As seen in Table 1, our calculated energy at
flare peak was about 8% higher than measured by GOES and thus we label this as an X27. If this
flare were to persist at peak level for a year, Table 1 indicates it would deliver about 171 kJ/m$^2$ to
the atmosphere. As shown by Brunton et al. (2023, their Figure 3), it is not uncommon for
supernova X-ray events to persist for over a year.  Table 1 shows that if our calculated X27 event
were to last about 2.3 years it would deliver about 400 kJ/m$^2$ which is the energy input postulated
by Brunton et al. (2023) as being biospherically destructive. Unfortunately, the problem with the
X27 simulation is that when this spectrum was input continuously into the atmospheric model
(TIME-GCM, discussed below), the model crashed after 8 days of the simulation. Thus, in our
discussion of ozone chemistry effects, we will discuss extrapolations based upon comparisons of
the nitric oxide response from the first 8 days of each simulation.
Figure 1 compares the spectra from our X13 and X27 calculations at their respective peak
minutes.  The figure shows the calculated spectrum at the native spectral resolution of
NRLFLARE (0.5 Å) and then integrated in 1 nm bins so that it can be compared to that derived
by Rodgers et al (2010, see their Figure 3). Like Rodgers et al. (2010), NRLFLARE shows a
significant increase in the flare spectrum from 1-2 nm relative to the shorter wavelengths less
than 1 nm.  As discussed by Siskind et al., (2022) this seems consistent with Orbiting Solar
Observatory (OSO) data presented by Neupert et al., (1967), although this spectral region is not
well covered with modern spectra. Comparing our results in detail with Rodgers et al. (2010),
suggests that our calculated $0.1 - 1$ nm flux of .004 W/m$^2$ is in good agreement. Our 1-2 nm
integrated energy is about 20% lower than Rodgers et al (2010) at flare peak. For the purposes of
this paper, this difference is not significant; when we compare our calculated nitric oxide
variation to Rodgers et al., (2010), we can account for this difference by using integrated energy
as the independent variable to normalize both our calculations. This will be discussed further in
Section 4.

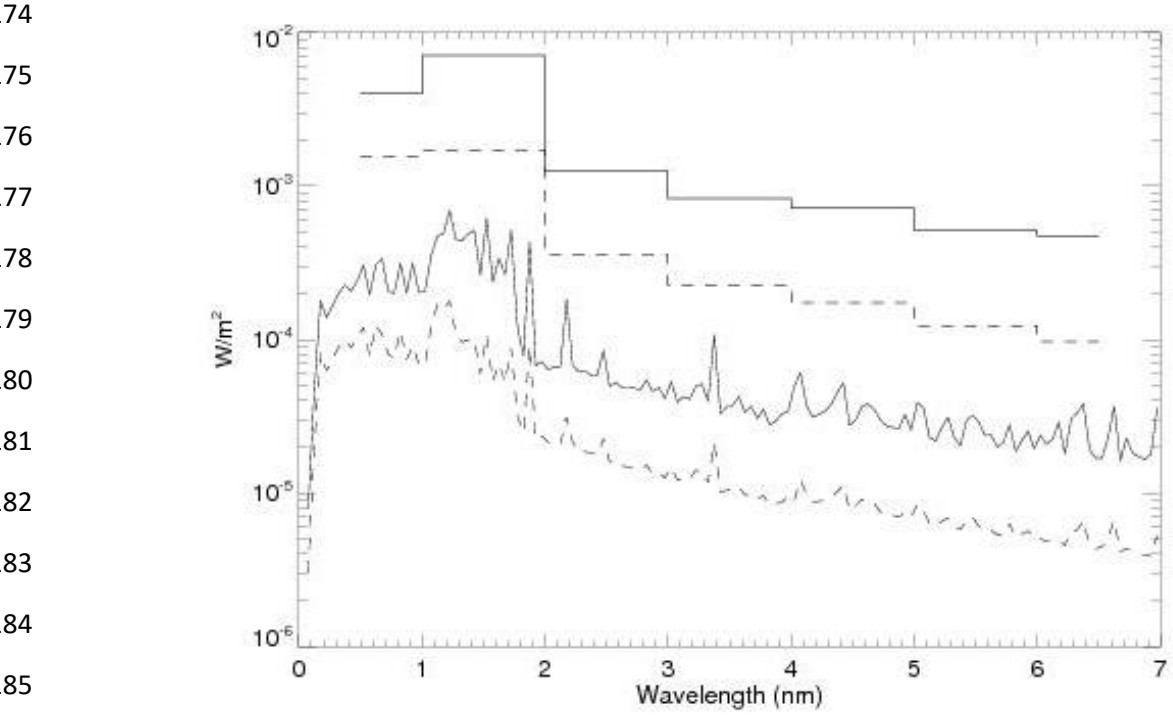

**Figure 1.** Calculated spectra for the peak of the X27 event of October 28, 2003 (solid lines) and the X13
event of Sept 10 2017. The rationale for the classifications is discussed in the text. The bottom two curves
are at 0.5 A resolution. The histogram format for the top two curves is the integrated energy over 1 nm
bins.

2.2. Atmospheric modeling with the TIME-GCM
The solar spectra shown in Figure 1 were used as inputs into the photoelectron ionization model
presented by Siskind et al., (2022) and incorporated into the NCAR TIME-GCM. The NCAR
TIME-GCM is a hydrostatic general circulation of the middle and upper atmosphere that solves
the continuity, electrodynamic, energy, and momentum equations from first principles on a
regular longitude and latitude and log pressure grid in the vertical (Roble and Ridley, 1994). The
model resolution is 2.5° x 2.5° (longitude x latitude) and 4 grid points per vertical scale height
extending from 12 to 4.6 x $10^{-6}$ hPa (or roughly 30 to 450-600 km depending on solar activity).
The photoelectron ionization model presented by Siskind et al., (2022) defines 12 new
wavelength bins for the soft X-ray energy range to give better spectral resolution (and hence
better altitude resolution of energy deposition) than the original NCAR spectral model presented
by Solomon and Qian (2005).  Note, there is a typographical error in Table 3 of Siskind et al.,
(2022), bin #7 for the $O_2$ cross section. It should read 1.5E-20, not 1.5E-21. It is correctly
implemented in the model.
One difference in how we used the TIME-GCM from the short term (< 1 day) simulations of
Siskind et al., (2022) concerns the dynamics of the mesosphere. In the standard version of the
TIME-GCM (i.e., the model setup used in Siskind et al., (2022)) climatological background
horizontal winds, temperatures, and geopotential are used at the model lower boundary in
combination with monthly mean diurnal and semidiurnal tides from the Global Scale Wave
Model (GSWM; Zhang et al., 2010a,b). However, this standard model configuration does not
properly simulate the downward transport of NOx from the mesosphere into the stratosphere. In
order to do so, we constrained TIME-GCM upper stratospheric and mesospheric horizontal
winds and temperatures between the model lower boundary (~30 km) and ~75 km with Modern
Era Retrospective-analysis for Research and Applications - version 2 (MERRA-2, Gelaro et al.,
2017) using four-dimensional tendency nudging (originally termed 4D data assimilation by
Stauffer and Seaman, (1990, 1994)). This nudging procedure is described in great detail by Jones
et al. (2018), and involves adding an additional acceleration and energy tendency term to the
conversation equations that is proportional to the modeled and MERRA-2 horizontal wind and
temperature differences up to ~75 km.
In previous studies (e.g., Jones et al., 2020; 2023), TIME-GCM was constrained using a high-
altitude version of the Navy Global Environmental Model (NAVGEM-HA, Eckermann et al.,
2018; McCormack et al., 2017), which provides dynamical fields up to ~97 km.  Note the
MERRA-2 reanalysis product used herein does not extend as high as NAVGEM-HA, and
therefore, we had to make a small modification to equation 5 of Jones et al. (2018). This equation
describes the vertical weighting distribution of nudging, which in part controls the strength of the
additional tendency term. The vertical weighting distribution used here takes the same functional
form as equation (5) of Jones et al. (2018), but the $z_{max}$ variable (representative of the TIME-
GCM log-pressure level where the model becomes unconstrained) is equal to -10.5 or ~75 km.
For reference, a vertical weighting factor of 0.5 occurs roughly at 55 km (or 0.2 hPa), above
(below) which the nudging term is more weighted toward TIME-GCM (MERRA-2) dynamical
fields. Finally, we conclude this section by noting that a key assumption that underlies our
approach is that all the enhanced NOx that comes flooding into the middle atmosphere would not
modify ozone so dramatically as to change the circulation away from that provided by MERRA-
2. Based upon the (small) degree of column ozone change shown below, we conclude this is an
acceptable assumption, but clearly could be investigated further with a more self-consistent
physical model.

2.3 Initial thermospheric response to multi-month solar flare
As discussed above, we model the effects of supernova induced soft X-ray event as if it were a
multi-month solar flare. Specifically, for the X13 event, we performed a simulation which
continues through the end of 2017 and then covers a complete additional year. In the analyses
discussed below, we present the results of the X13 and X27 simulations with a baseline run that
only includes the EPP-IE effect. The difference between the X13 or X27 and baseline runs serve
to quantify the possible response of the middle and upper atmosphere to a multi-month soft X-
ray event. We also note that for TIME-GCM simulations performed herein geomagnetic activity
was held constant with Kp $\cong$ 3 in order to exclusively highlight flare impacts.
Figure 2 shows the initial response at low latitudes (averaged from 30S-30N), plotted every two
hours, as a function of longitude for the first day. The solid line is 1600 UT which was just at
flare onset (the peak of the Sept 10, 2017 flare was around 1606 UT). The four dashed lines are
for 1700, 1900, 2100 and 2300 UT and show how the NO increases both in the thermosphere
(panel (a)) and in the mesosphere (panel (b)) immediately after flare onset.  Note how the
longitudinal response progresses westward for the equatorial plots, tracking the sub-solar point.
This is consistent with our implicit assumption that the supernova will be aligned with the
ecliptic plane. While perhaps not always true (the galactic plane is tilted 60$^{\circ}$ with respect to the
ecliptic plane (cf. https://en.wikipedia.org/wiki/Astronomical_coordinate_system), any
supernova will nonetheless rise and set like the sun, and the peak effects will, like with a solar
flare, be concentrated at the sub-stellar longitude. Thus we conclude that our approach of using
an extended solar flare event as a means of simulating a supernova soft X-ray event is
acceptable. In our conclusions, we will discuss the possibility of a high latitude supernova soft
X-ray event further.
Figure 3 shows daily averaged profiles for the first 10 days for the event, both at low and at high
latitudes. The effects are largest at the equator, but are still significant at 59S, and extend well
down into the mesosphere. Note the changes appear to level off after several days, suggesting
that the initial response is saturating. Indeed we find that all the thermospheric response occurs in
the first 10-14 days. The middle atmosphere response includes both this initial effect and then
later, seasonal effects as NO is transported down from the upper mesosphere/lower
thermosphere.

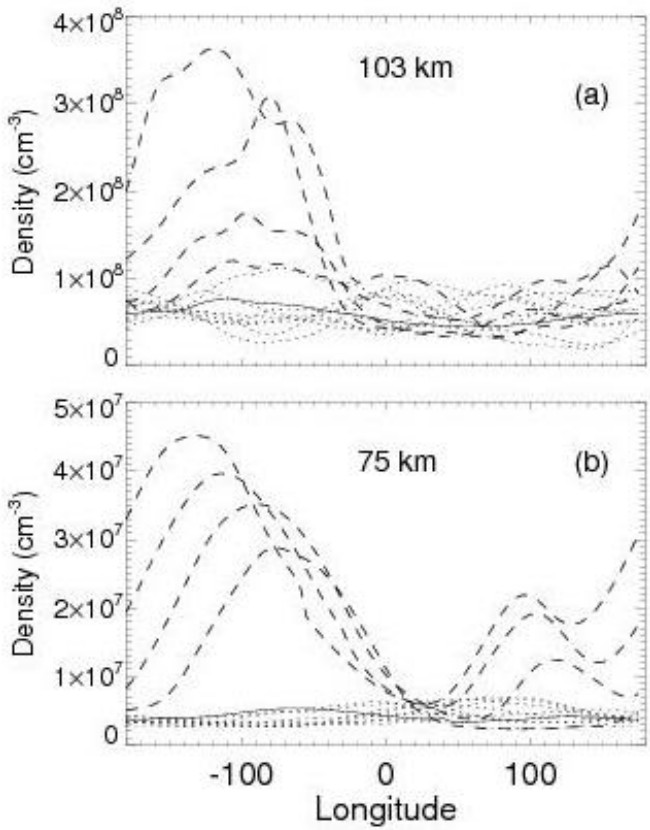


**Figure 2**. Initial response of thermospheric (panel **(a)**) and mesospheric (panel **(b)**) nitric oxide density to the onset of the extended flare. The solid line in each panel is for 1600 UT, which roughly corresponds to the onset of the flare. The dotted lines are for times prior to that. The dashed curves which progressively increase and phase to the left according to the sub-solar point are for hours 1700, 1900, 2100 and 2300 UT.



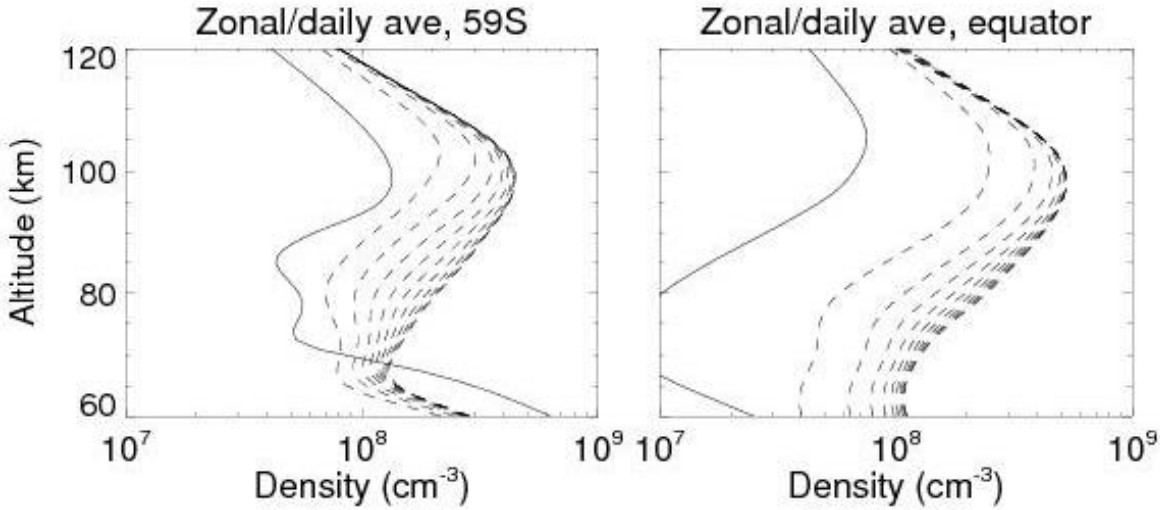


**Figure 3.** Profiles of the first 10 days of the nitric oxide profile at two latitudes. The individual days are not labeled, but the day-to-day increase in NO density is monotonic with time. The solid lines are pre-flare.


## 3. Seasonal Variation of the Xray-IE in the middle atmosphere

In order to provide a broad, but quantitative, overview of the production of NOx from the extended flare/supernova, Figure 4 shows the calculated total number of NOx molecules in units of gigamoles (GM) and compares it to a baseline/no flare simulation. This quantity has been previously used (Vitt and Jackman, 1996; Siskind et al., 2000; Funke et al., 2005) as a way of quantifying space weather impacts on the ambient NOx budget. Here, the production of NOx is mostly in the mesosphere while the impacts on ozone are in the stratosphere. Therefore, using the 50 km level as an arbitrary dividing line, we break out our calculation to illustrate mesospheric NOx (top panel of Figure 4) and stratospheric NOx (bottom panel of Figure 4) separately.

In each panel, the upper (solid) curve is the NOx with the extended flare calculation. The dashed curve is a baseline case with no flare. First, considering the no flare case, our stratospheric value equilibrates to around 20-22 GM (we attribute the initial decrease to an excess of NOx in the initial conditions). Given that the model bottom boundary is 30 km and that significant NOx lies below 30 km, our result is likely consistent with previous estimates by Vitt and Jackman (1996) of 29-30 GM for the stratospheric production of NOx from $N_2O$ oxidation. For the no flare case, the upper panel shows a value between 3-5.5 GM due to the background secondary NOx maximum in the upper mesosphere/lower thermosphere.

For the flare case, the mesospheric results show a rapid increase to over 15 GMs. The stratospheric NOx does not increase immediately, but as evidenced by the increasing divergence between solid and dashed curves, shows a gradual increase in the flare produced NOx. It is

interesting that for all 4 curves, the maximum NOx occurs in the period from days 570-620. This
corresponds to August and September and coincides with the late winter period in the Southern
Hemisphere. As we will discuss, satellite analyses have indicated that the maximum delivery of
upper mesospheric/lower thermospheric NOx to the stratosphere occurs during that time and, as
we show below, this is indeed the case here.

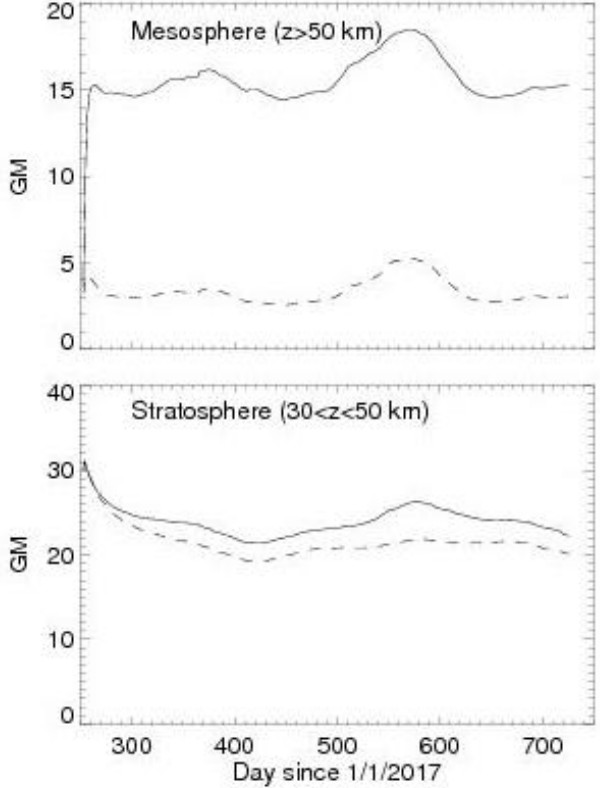


**Figure 4. Total globally integrated NOx (=NO + NO2) number of molecules (GM: gigamoles) for**
**the baseline no flare case (dashed line) and the continuous soft X-ray flare (solid line) for the**
**mesosphere (top panel) and stratosphere (bottom). The soft X-ray event, which assumes a spectrum**
**from the Sept 10, 2017 flare, is assumed to have begun on that day (day 253 of 2017).**


Finally, we can give a crude comparison of the global effects of this extended flare to previous
space weather phenomena. The largest difference in the stratosphere between the flare and
baseline, as shown in the bottom panel of Figure 4, is ~4.5 GM. This can be compared to the 1.3
GM that Funke et al., (2005) estimated was delivered to the upper stratosphere during the 2003
Antarctic winter which followed a period of elevated space weather activity. Thus the extended
flare appears to exceed that by about a factor of 3.5. Funke et al., (2005) also estimated a
roughly equivalent amount of NOx would end up in the lower stratospheric polar vortex, below
our 30 km bottom boundary. Siskind et al., (2000) also estimated a peak vortex amount of about
0.8-1.3 GM. If we assume this rough equivalence between upper stratospheric and lower
stratospheric polar vortex delivery applies here, then we arrive at an estimate of 9 GM from this
extended X13 flare. By comparison, Vitt and Jackman (1986) estimated a total production of 7
GM from the large solar proton event in 1989. Thus our current simulation exceeds any
previously documented space weather effect on stratospheric NOx, but at the same time, it is not
dramatically bigger. As we shall see when we look at the details of the NOx distribution and its
effects on ozone, our results follow that pattern i.e., greater, but not dramatically so.
Figure 5 compares the seasonal variation of the TIME-GCM NOx (defined as $NO + NO_2$) from
our extended flare calculation with our baseline run that only includes the EPP-IE. It thus shows
the seasonal variation of how the Xray-IE leads to NOx buildup in the middle atmosphere
beyond that caused by energetic electron precipitation. To understand this, we first focus on our
baseline EPP-IE simulation and how it compares with the recent simulations of the EPP-IE from
Pettit et al., (2021), specifically their Figures 9-10 which they compared with Michelson
Interferometer for Passive Sounding (MIPAS) data in the Southern Hemisphere. Ultimately, we
will conclude that the Xray-IE shows similar behavior to the EPP-IE simulation, except with a
larger magnitude and for a more prolonged seasonal duration. Thus to highlight the longer
impact, we show the entire year whereas Pettit et al., (2021) just showed April-October.
In comparing with Pettit's results, we see that our baseline simulation underestimates the descent
of the MIPAS NOx data at the higher latitudes. The MIPAS data show the 16 ppbv contour
descending to below 35 km for the month of August, whereas our simulation (panel a) has this
contour remaining above 40 km for the late austral winter period. There are likely two reasons
for this. First, is likely the simple fact that TIME-GCM has a bottom boundary at 30 km and thus
the descent will decay as this boundary is approached. Indeed, analyses of data from both the
Halogen Occultation Experiment (HALOE) on board the Upper Atmospheric Research Satellite
(UARS)and Polar Orbiting Aerosol Measurement (POAM) data have shown that enhanced NOx
can routinely be detected below 30 km in the Southern Hemisphere (Siskind et al., 2000; Randall
et al., 2007). Second, our model does not have the medium energy electron ionization that Pettit
et al., (2021) discuss. They show that models without this component of energetic electrons
underestimate the descent of NOx into the mid-stratosphere.
On the other hand, our baseline simulation does much better at mid-latitudes (38-53S in the
figure). It shows the 16 ppbv contour dipping down to 45 km for a couple of months. This is
quite similar to the MIPAS data shown by Pettit et al., (2021) and is consistent with Funke et al
(2005) and Arnone and Hauchecorne (2011) who pointed out that there are two components to
the descent of upper atmospheric NOx into the stratosphere. One component is directly into the
stratospheric polar vortex and descends down into the mid-stratosphere; as we note above, our
model cannot capture this. However, there is a second component that is dispersed into middle
latitudes in the upper stratosphere. It appears that our model does capture this and it could be
argued that from a global biospheric perspective, this second component is more important since
a greater region of the globe is affected.

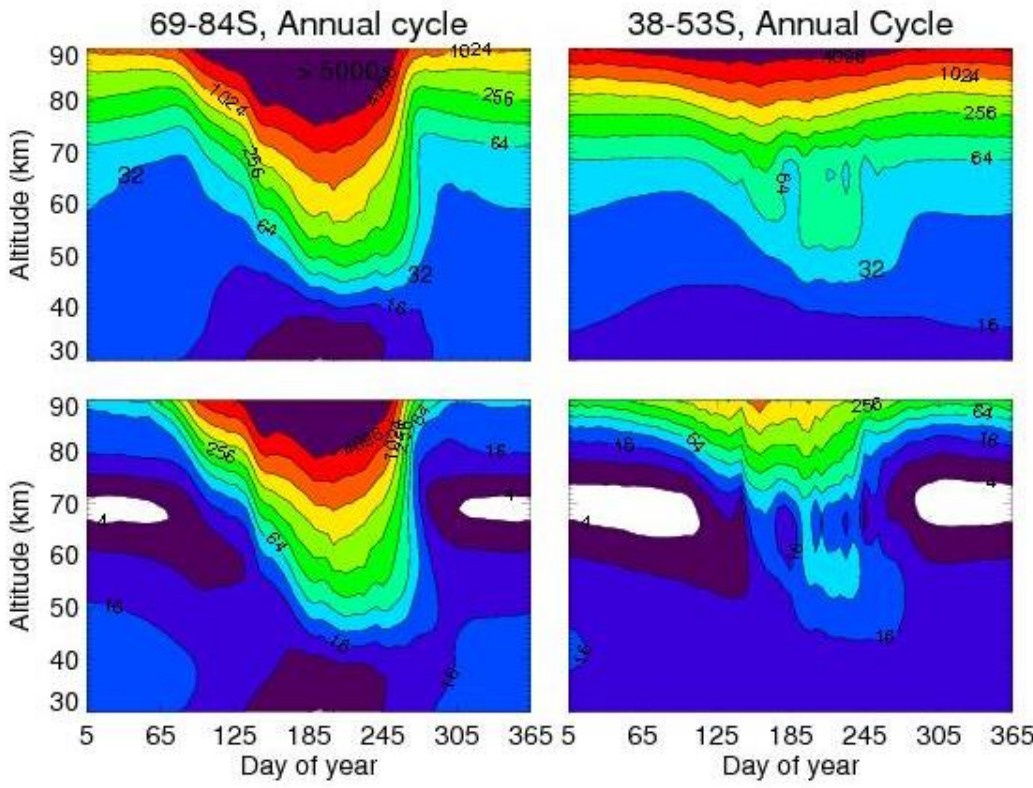


**Figure 5.** Annual cycle of NOx descent into the upper stratosphere from TIME-GCM for two latitude
bands. The bottom row is for a baseline simulation that only includes the EPP-IE. The top row
additionally includes the Xray-IE from the X13 simulation presented in Figures 1-3. The year shown is
2018 thus representing the period about 4-12 months after flare onset on Sept, 10, 2017. The values on the
contour labels are in units of ppbv. The white colored regions in the baseline run are for mixing ratios < 4
ppbv.

Regarding our Xray-IE simulation, dramatic effects are clearly seen in the mesosphere, both mid
and high latitudes. The mesospheric minima near 70 km are completely filled in and mixing
ratios of over 32 ppbv, up to near 100 ppbv, are seen for most of the year. However, for
considerations of impacts on ozone, we focus more on the stratospheric effects. Here, at first
glance, for the higher latitudes, the IE-Xray effect appears somewhat muted. We see no
difference in the maximum value of NOx descending below 50 km between our baseline and
constant X13 simulation. However, the IE-Xray effect is somewhat more prolonged in its NOx
enhancement. The baseline simulation shows the 16 ppbv contour curving sharply upward
around Day 270. Thus NOx values near 50 km decrease abruptly and this is similar to what is
seen in Pettit et al.'s MIPAS data. However, the X13 simulation shows the upper stratospheric
NOx values remaining between 16-32 ppbv for the entire austral spring.

At mid-latitudes, the effect of the continual soft X-ray flux is more pronounced. Whereas the
baseline simulation shows 16 ppbv descending to about 45 km, the flare simulation has about
double that.  Like the high latitude case, after approximately Day 270, the baseline case NOx
values fall below 16 ppbv, in agreement with the MIPAS data. By contrast, in the X13

simulation we see NOx values of 32-64 ppbv descending to 45-50 km and the entire upper
stratosphere remains flooded with enhanced NOx values greater than 16 ppbv for the whole year.
Figure 6 also compares our baseline (EPP-IE only) simulation with that including the Xray-IE,
this time for two pressure surfaces as a function of latitude and time: one near the stratopause
(the indicated pressure roughly corresponds to altitudes of 45-48 km) and one lower down
towards the middle stratosphere (approximately 38-40 km). The figure shows how the NOx from
the flare/supernova spreads over the Southern Hemisphere. It is useful to first look at our
baseline case; it clearly shows that the EPP-IE effect is mainly in late winter/early spring in the
Southern Hemisphere and covers the latitudes from -80 to about -20 or -30. Note, there is no
evidence for this seen at 3.0 hPa whereas in actuality, there should still be a spring time
enhancement in the highest latitudes as we discussed above. When we compare this with the top
row in the figure, the effects of the soft X-rays are very apparent. The late winter/spring
enhancement at 1.1 hPa is about twice as large and there is now seen an enhancement at 3.0 hPa
whereby values of NOx of 10-12 ppbv at Southern mid-latitudes are now replaced by values of
14-16 ppbv. Importantly, there is no evidence for significant enhancements in the Northern
Hemisphere although there does seem to be a general global increase in NOx of about 2 ppbv-
about 20% above the baseline values. This lack of significant NH enhancement is consistent with
observations of the EPP-IE which show generally weaker effects in the NH relative to the SH
(Funke et al., 2014). This is generally believed to be due to the weaker descent in the NH and the
greater horizontal mixing due to mesospheric planetary waves (Siskind et al., 1997), although
NH enhancements are seen in specific years with very strong dynamical perturbations (cf. Funke
et al., 2017). In the present case, while we will consider the effects on stratospheric ozone below,
it does suggest a limit as to how biospherically destructive the soft X-ray event could be since
the effects are likely to be much more muted in the NH
One final consideration in looking at the annual cycles in the upper stratosphere mesosphere in
Figures 5 and 6 is that there appears to be no evidence for any continual buildup of NOx. The
NOx at the end of 2018 is not much different than at the beginning. This is consistent with
Figure 3 in that the day-to-day NO increase in the thermosphere decreases such that after 10 days
the NO profile showed little change. This will be important when we try to extrapolate from our
X13 simulation to stronger events.
Figure 7 shows the global change in ozone for the X13 simulation compared with our baseline
EPP-IE only case for four pressure surfaces ranging from 0.68 to 3.0 hPa. The ratios are less than
1.0 globally for the entire year which means lower ozone for the X13 simulation. However, there
is a clear maximum in the reduction for the late winter/early spring period in the SH, consistent
with the global distribution of the enhanced NOx shown in Figure 5. Note that the fractional
reduction is larger at the lowest pressures. Normally, at these altitudes in the lower mesosphere,
ozone loss is dominated by the HOx catalytic cycle (Brasseur and Solomon, 2005). However,
with NOx enhancements on the order of 100 ppbv, the NOx catalytic cycle can dominate up to
higher altitudes (lower pressures) than is conventional. At the same time, since the bulk of the
ozone density is in the stratosphere, the effect of a 3-4% reduction at 3.0 hPa is of greater impact
than a 10% reduction at 0.68 hPa.

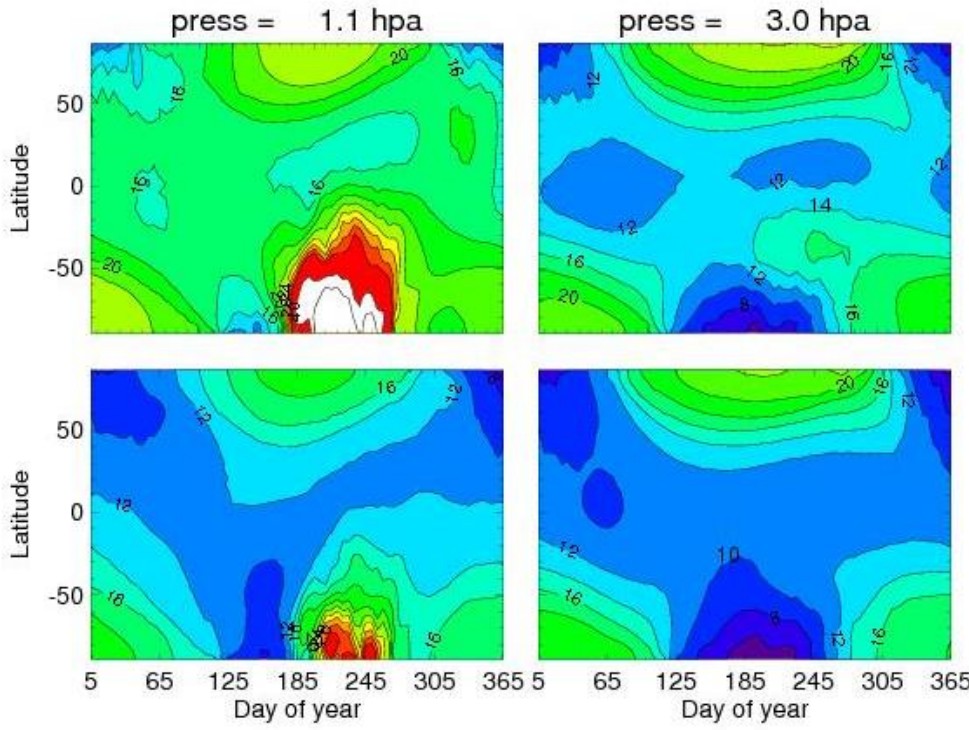


**Figure 6.** NOx (ppbv) vs latitude and day of year. The period of time is the same as shown in Figure 4. The bottom row is for the baseline case without enhanced soft X-rays; the top row includes the continuous X13 flux. The red regions are NOx values greater than 28 ppbv; the white regions are NOx values greater than 40 ppbv.

433

The results show here clearly suggest a potentially global effect on the ozone, albeit limited to a couple of months when the SH NOx enhancement has spread to the equator. The effect is not large- about 5% locally in the upper stratosphere and thus unlikely to be biospherically significant. However, there are important caveats to this statement that we will explore in the subsequent section. First, as we noted above, our input X-ray energy is much smaller than the supernova soft X-ray events postulated by Brunton et al., (2023). Second, the TIME-GCM is limited by a bottom boundary at 30 km. About half of the stratospheric ozone column lies below this altitude and must be considered before drawing any conclusions. We consider both these issues in the sections below.

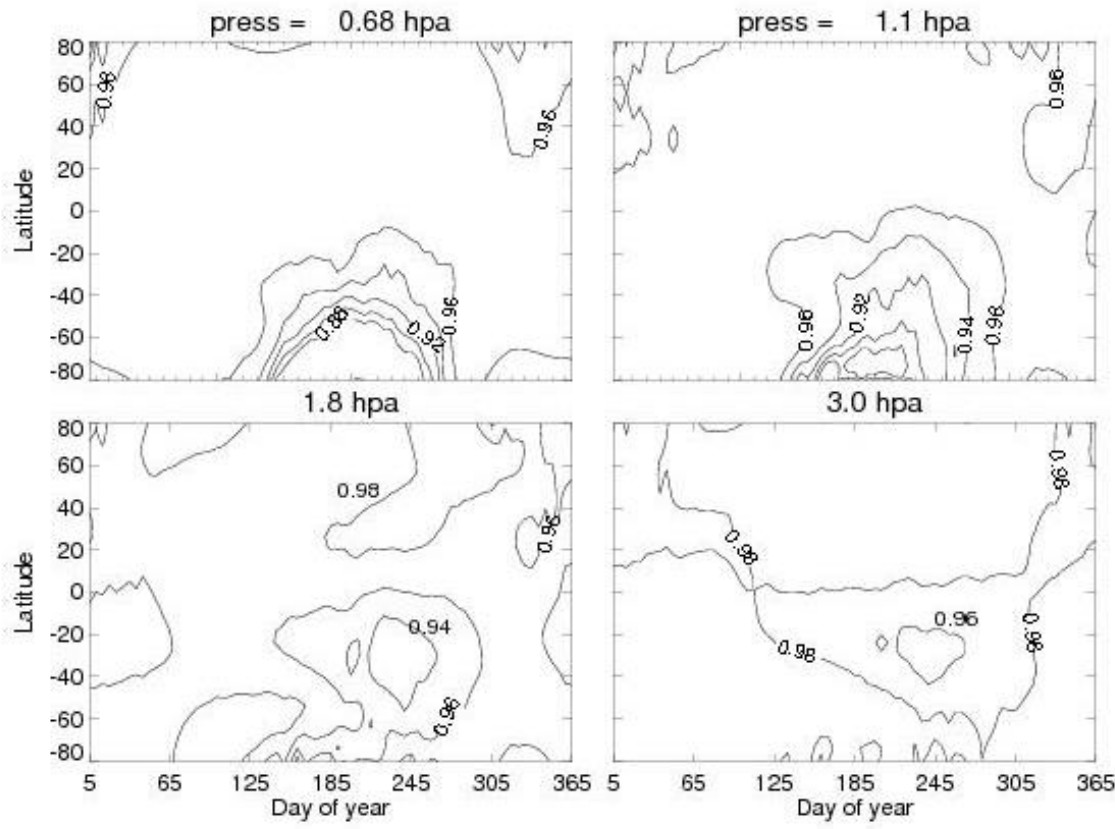

443

**Figure 7**. Annual variation of the ratio of ozone from the X13 simulation compared with the baseline
simulation at the four indicated pressure surfaces

446

## 4. Extrapolation to higher X-ray fluxes and impact on stratospheric ozone

To extrapolate our NO/flare response, we first seek to compare our results with observations of
the NO response to solar flares. The only quantitative data analysis of the response of nitric
oxide to a solar flare that we are aware of is that by Rodgers et al. (2010) using data from the
Student Nitric Oxide Explorer (SNOE). SNOE was particularly well suited to study the NO
response to a solar flare because it was in a sun-synchronous orbit with an equator crossing time
in the late morning when the sun was relatively high in the sky. Rodgers et al. calculated the NO
column change observed by SNOE and plotted it versus the integrated soft X-ray input energy
derived from a catalog of 11 flares.

Figure 8 compares the TIME-GCM results to Rodgers. The figure shows the integrated energy
from the four strongest X-class flares observed by SNOE with the largest being the so-called
Halloween event of October 28, 2003. As noted above, this event, labeled as X18 in Rodgers et
al.'s Table 3, is now recalibrated to be X25, and in our simulation with NRLFLARE it is a bit
higher at X27. Also shown are the TIME-GCM calculated hourly column NO from the local

equatorial sub-solar longitude for each of the first 24 hours of our model simulations for the X13
and X27 events.

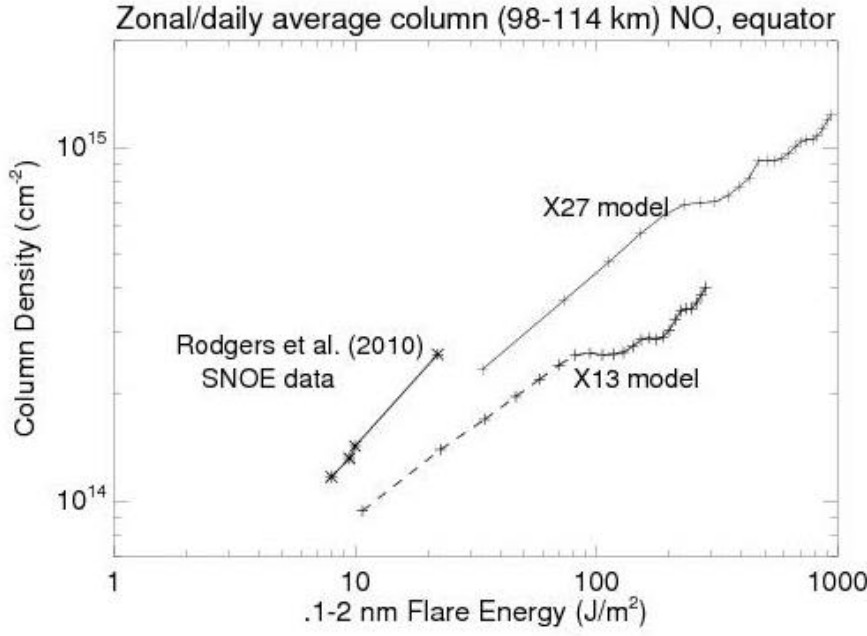


**Figure 8.** Calculated TIME-GCM NO column density enhancement from the X13 and X27 simulations
compared with the observed NO increases reported by Rodgers et al. (2010) for the 4 strongest flares
listed in their Table 3. The plus symbols on the model curves represent output for every hour. The first
points shown for each of the model account for the number of minutes after each integral hour that the
flare peaked. Thus the X13 flare peak was at 16.1 UT (cf. Table 1 of Siskind et al., 2022) and thus the
first point shown for the X13 model represents 54 minutes of photon flux. Like Rodgers et al. (2010), we
subtracted the pre-flare NO column in the model before calculating the enhancements shown.

In general, the figure shows a quasi-linear relationship between column NO and the integrated
energy for both SNOE and the two model simulations. It appears that the rate of energy input is
important for the NO increase. Thus after two model hours, the X13 simulation accrues the same
energy input as the 27 minute long October 28, 2003 flare and yet the NO column response is
well below the observations. The column NO for the X13 simulation takes over 4X the energy
input of the observed flare to reach the same enhancement as observed by SNOE. The column
NO for our X27 simulation, which is designed to simulate the October 28, 2003 flare comes
closer and matches the SNOE data just after the first hour of the model simulation (actually 51
minutes since the flare peak was at 9 minutes past 11 UT and model output was only saved
hourly). However, since the actual October 28 flare only lasted 27 minutes, it means that the
TIME-GCM is calculating a smaller NO column for the same energy input than was recorded by
SNOE. Rodgers et al. (2010) reported an observed column enhancement of 2.6E14 cm$^{-2}$ for solar
X-ray input of 22.4 J/m$^2$ where, reading from the graph, the TIME-GCM requires closer to 40
J/m$^2$ before reaching this level of NO enhancement.
After 24 hours, Figure 8 shows that the X27 simulation produces about a factor of 3 more NO
than the X13 simulation. Figure 9 shows the daily averaged, zonal mean column NO for both
models extended out to the full 8 days of the X27 simulation before the model crashed. Similar
to Figure 3, it shows that both models level out after several days. The ratio of the two column
densities equilibrates to a slightly smaller value than seen in Figure 8, about a factor of 2.6. The
fact that the column densities level out can offer a useful guide for extrapolating our middle
atmosphere NOx enhancements even without completing a full year with the X27 simulation. It
suggests that the reasonable enhancements might lie in the range of a factor of 2-3 over the X13
simulation. In terms of GM as presented in Figure 4, it may suggest a net delivery to the
stratosphere of 20-26 GM for the X27 case. We will consider the consequences of this below.

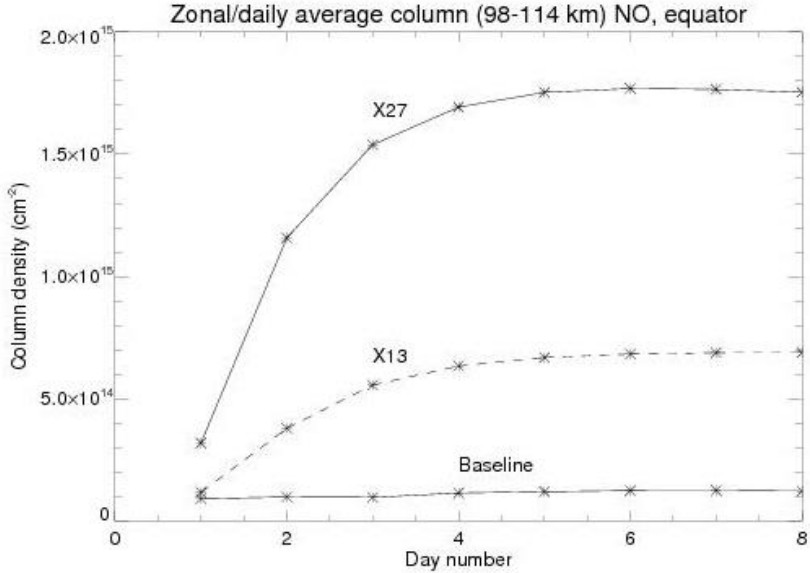


**Figure 9.** Daily and zonally averaged equatorial column densities for the X27 (solid line with stars) and
X13 (dashed line with stars) TIME-GCM simulations. A baseline case run for the conditions of
September 2017, but with no flare/supernova and which remains at approximately $1 \times 10^{14}$ cm$^{-2}$ is also
shown.
To evaluate in detail how ozone may be reduced for the X27 simulation, we will use the
CHEM1D photochemical box model. This model has previously been used to model satellite
observations of mesospheric OH (Siskind et al., 2013) and validate ground-based measurements
of ClO (Nedoluha et al., 2020). It is important to first evaluate the model's ability to calculate
stratospheric ozone since, as is most recently discussed by Diouf et al. (2024), chemical models
of upper stratospheric and lower mesospheric ozone historically fall short of fully reproducing
observations.
Figure 10 shows a comparison of CHEM1D and TIME-GCM ozone with two observations from
September 2nd, (Day of year 245) 2018 at a latitude of 38-40S. This period and location was
selected because it corresponds to the time and location of the most significant upper
stratospheric ozone depletions indicated by the TIME-GCM in Figure 6. The observations are
from the 9.6 $\mu$m measurement of the Sounding of the Atmosphere with Broadband Emission
Radiometry (SABER) instrument on board the NASA TIMED satellite and the Microwave Limb
Sounder (MLS) from the NASA Aura satellite. SABER and MLS data have long been the
standards for measuring middle atmospheric ozone globally. Figure 10 shows, first, that TIME-
GCM is ill suited for model-data comparisons of stratospheric ozone. This is perhaps not a
surprise- the model was designed to study middle atmospheric dynamics and transport and its
coupling to the upper atmosphere (Roble et al., 1994). For example, TIME-GCM does not
include all the active chlorine and nitrogen species that are required for a comprehensive model
of stratospheric ozone. Thus for chlorine, TIME-GCM has Cl and ClO, but not HOCl. For
nitrogen, TIME-GCM only has NO and $NO_2$, but not $HNO_3$ or $N_2O_5$. By contrast, CHEM1D
does include these species. The comparison with CHEM1D very closely matches that seen by
Siskind et al. (2013), who used CHEM1D for mesospheric ozone and hydroxyl and Diouf et al.
(2024), who used the model of Bertaux et al. (2020) and compared with MLS ozone and SABER
$O_2(^1\Delta)$ 1.27 $\mu$m emission. In all cases, the model falls short of completely reproducing the
observations. Both Siskind et al. (2013) and Diouf et al. (2024), having exhausted all possibilities
for reaction rate changes and possible temperature inputs, invoked the possibility of an additional
source of ozone from vibrationally excited oxygen as hypothesized by Slanger et al., (1988) and
Price et al., (1993). The purpose here is not to answer this long-standing question; rather, Figure
10 shows that CHEM1D does as well as could be expected given our understanding of middle
atmospheric ozone photochemistry. Our purpose here is to perform sensitivity studies for varying
amounts of NOx, guided by our TIME-GCM simulations. Figure 10 shows that CHEM1D is
adequate for this task. We should additionally note that as one moves towards higher pressures
greater than 5 hPa, the chemical lifetime of ozone becomes longer such that it is no longer under
pure chemical control but also dynamical influences. Thus, the apparent improved agreement
with the observations near 10 hPa should not be over-interpreted.

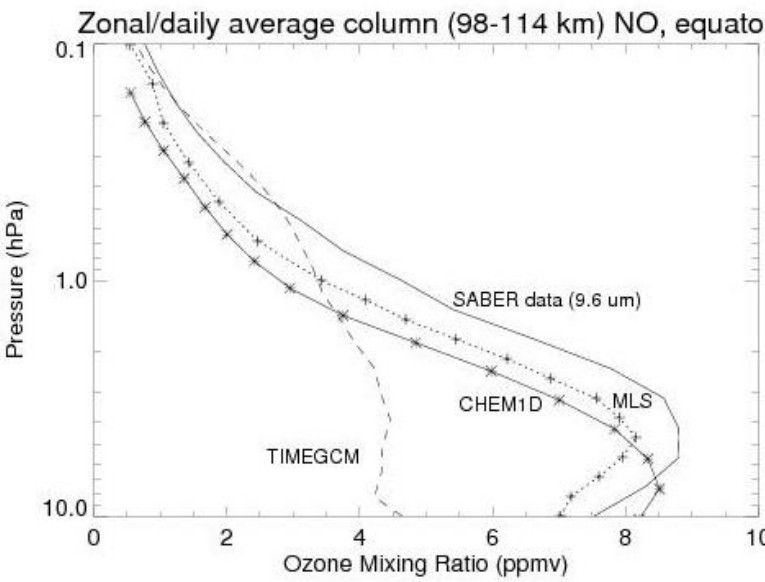


**Figure 10.** Comparison of the TIME-GCM (long dashes) and CHEM1D (solid line with stars) models
with SABER (solid line) and MLS (dotted line with plus symbols) observations of ozone. The location is
38-40S and the time of year is September 2[nd], 2018. CHEM1D used temperature and pressure and NOx
abundances from the TIME-GCM as input. The approximate altitude range corresponding to the y-axis is
about 30-62 km.
We now show the fractional ozone depletions, as a function of pressure, from the enhanced NOx
due to a multi-month solar flare. Figure 11 presents the calculated ozone loss ratios (panel a) for
two models of CHEM1D that use enhanced NOx compared with the baseline simulation
presented in Figure 10. The location and time of year is the same as in Figure 10. The NOx
enhancements (panel b) are taken from the X13 simulation shown in the previous figures plus an
extrapolated enhancement (the greater of the curves in Figure 10) based upon the short-term
response shown in Figure 9. Figure 11 also shows the vertical profile of the TIME-GCM ozone
change taken from Figure 7.

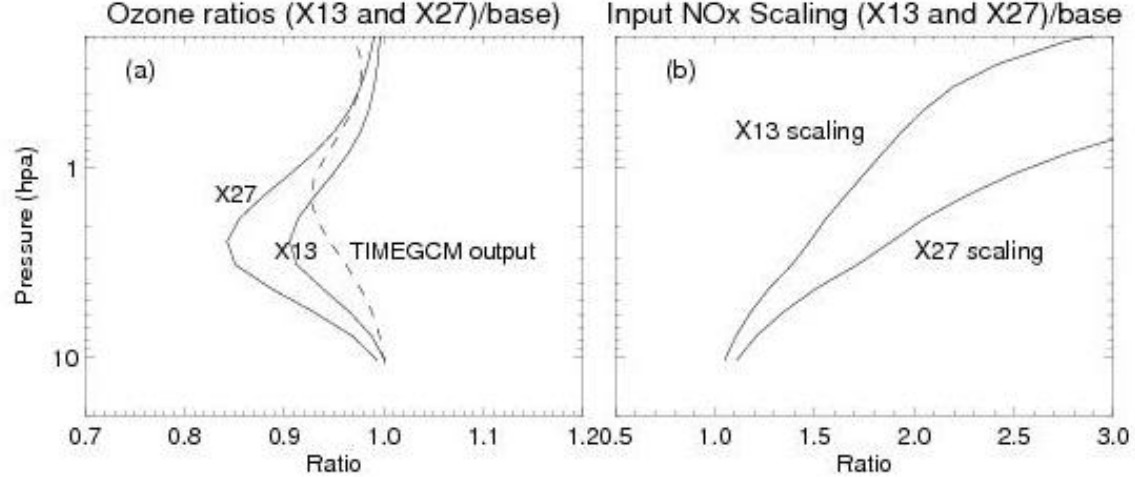


**Figure 11. (a)** Ratios of calculated ozone from CHEM1D compared with a baseline (no flare) case for
September 2, at a latitude of 39S.  The two solid lines use NOx input according to the scaling ratios
shown in panel **(b)** The X13 scaling is based upon the NOx shown in Figures 3-5. The X27 scaling is a
hypothesized extrapolation based upon Figure 8 and discussed in the text.  Also shown as the dashed line
in panel (a) is the ozone ratio from the TIME-GCM  as per the surface contour plots shown in Figure 6.

Figure 11 shows that for the X13 case, we could expect ozone depletions of up to 8% in the
upper stratosphere. For the more significant X27 case (i.e., for a more intense supernova X-ray
event), we might see ozone reductions of up to 15-18% in the upper stratosphere.  Figure 11 also
shows the vertical profile of the TIME-GCM ozone reduction. It does not exactly match the
profiles from CHEM1D in terms of shape and altitude of peak reduction, but it is very close to
the X13 CHEM1D simulation in terms of giving a peak loss of 6-7% in the upper stratosphere.
The TIME-GCM result is useful because it allows our detailed CHEM1D calculations to be
placed in the global context shown in Figure 7.
Based upon Figure 11 and Figure 7, we can conclude that a supernova soft X-ray event could
cause widespread ozone loss in the 10-20% range in the upper stratosphere for late winter/early
spring in the Southern Hemisphere. While this would likely be easily observable with suitable
instrumentation, it is less likely to have a dramatic biospheric effect. This is because most of the
stratospheric ozone is found at altitudes from 20-35 km (5 hPa-50 hPa pressure levels). The
losses shown in Figure 11 are only the upper edge of that layer. This is shown in Figure 12,
which shows the actual ozone mixing ratios (panel (a)) and ozone density profiles (panel (b))
which correspond to the scaling ratios shown in Figure 10. In the case where the model output is
shown as ozone densities, the curves are almost indistinguishable. The change in the total
column ozone, which is most relevant for surface UV exposure, is 1% for the X13 simulation
and 2% for the X27 extrapolation.

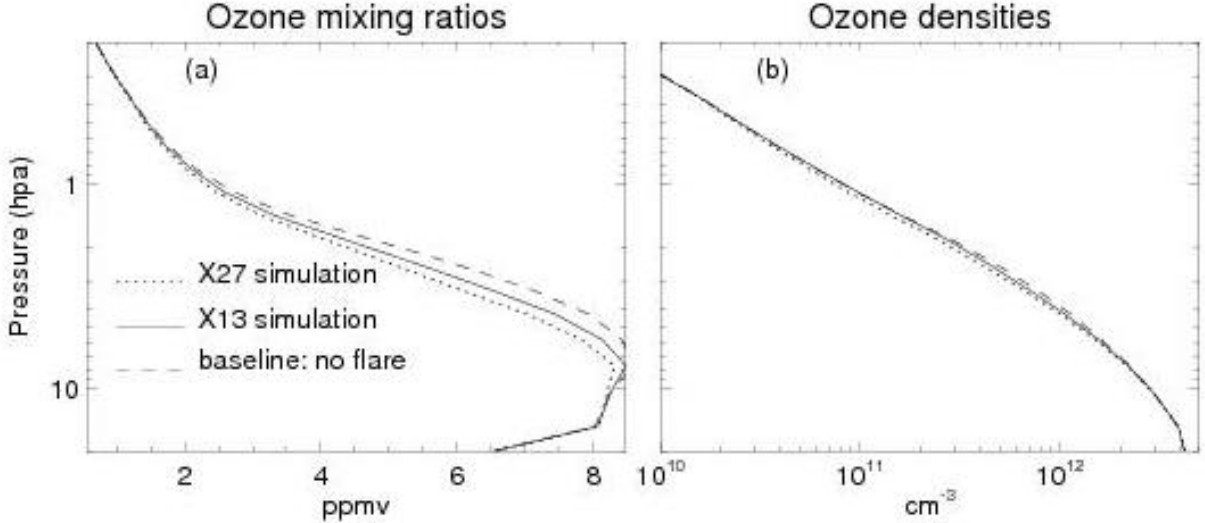


**Figure 12.** Absolute ozone abundances corresponding to the ratios presented in Figure 10. The three
simulations are labeled in panel (**a**). They are identically shown in density units in Panel (**b**) but are
almost indistinguishable because the 8-15% reductions are very hard to see on a graph that covers over
two orders of magnitude.

## 5. Discussion and conclusions

Our results clearly suggest the strong possibility of globally widespread ozone loss in the upper
stratosphere, at least for a period of a couple of months in the Southern Hemisphere. However, at
the same time, we conclude that this is unlikely to have a global biospheric impact because the
depletion is limited to the upper edges of the ozone layer. This limitation is derived from our
simulations showing that, like the EPP-IE, the Xray-IE does not penetrate below 35-40 km on a
global basis. At polar latitudes, our results allow us to speculate that a supernova could greatly
exacerbate the ozone hole. Or even, for atmospheres without anthropogenic chlorine, create an
ozone hole. Indeed, it has already been noted that the EPP-IE has been confused with an

expansion of the ozone hole due to volcanic aerosols (cf. Siskind et al., 2000 and discussion therein). However, since the hole is generally confined to the polar vortex, the effects of the Antarctic ozone hole have not caused widespread global ecological destruction although regional effects may be occurring (Robinson et al., 2024). There are likely other more subtle hypothesized effects of the enhanced NOx that we do not address. For example, we do see moderate NOx enhancements throughout the Northern Hemisphere and it has been suggested that EPP-IE in the Northern Hemisphere has effects on stratospheric and possibly tropospheric meteorology (Seppala et al., 2009). Our work here cannot rule this out for the Xray-IE.

Certainly, our results come with large uncertainties that would be useful to address. Perhaps the biggest is that the TIME-GCM, with a bottom boundary above the peak of the ozone layer, is not designed to study stratospheric chemistry. Moreover, the 30 km bottom boundary prevents us from studying descent of NOx enriched air down to the lower altitudes where the EPP-IE has been observed in the SH polar vortex (Randall et al., 2007). Thus our comments about the ozone hole are necessarily speculative. In addition, our simulation of the NO produced during solar flares appears to be less than observed by SNOE. This might mean that the NO response to a flare would be greater than we suggest, perhaps by as much as a factor of 2. Here it would be very helpful if there were another dataset that could corroborate the NO response reported by Rodgers et al., (2010). As we noted above, the local time of the sun-synchronous SNOE orbit was ideal for observing solar flares. By contrast, more recent NO observations which are summarized in Table 1 and Figure 3 of Emmert et al., (2022) are less well suited. Emmert et al. (2022) show that, for example, the Atmospheric Chemistry Experiment (ACE) and the Solar Occultation for Ice Experiment (SOFIE) on the NASA/AIM satellite used the technique of solar occultation which by definition means sunrise or sunset. This type of observation is not well suited to observing the effect from a flare that would be less noticeable at local sunset or sunrise. Likewise, the ODIN satellite, which measured NO with the Sub-millimeter radiometer (SMR) was in a dawn-dusk synchronous orbit. Based upon Emmert et al., (2022) it appears that only MIPAS on the ENVISAT satellite was in a proper daytime orbit to see flares. An examination of the MIPAS data might be an interesting test of some of our SNOE-based results.

Ultimately, however, even if we did underestimate the NO production by a factor of 2 or even 3, the effects on the ozone column are likely not catastrophic because they will be limited to above 35-40 km. We point to the simulations of Thomas et al., (2007) of a possible solar proton event that may have accompanied the 1859 Carrington flare event. Solar protons penetrate much deeper into the stratosphere than soft X-rays and thus the effect on NOx is more direct rather than indirect as simulated here. Indeed, they obtained much larger NOx increases down to 30 km and localized ozone losses near 35-40 km of greater than 30%. Despite this greater increase in NOx and greater ozone loss, their calculated perturbation to the ozone column was less than 15% because the bulk of the ozone density between 20-30 km remained unaffected from the proton flux. More recently, Reddman et al., (2023) performed a similar simulation of an extreme solar proton event combined with an extreme geomagnetic storm. They show dramatically enhanced ionization in the high latitude regions for all altitudes above 30 km. Their extrapolated NOx production is approximately 25-30 GM roughly equivalent to our extrapolation for our X27 case, but now occurring directly at higher latitudes where transport to the lower stratosphere might be

hypothesized as more efficient. However, like our results, they find the overall impact of any
resulting ozone reduction on UV flux to the surface to be limited to less than 5%. The Reddman
simulation is important because it might be relevant to the question of whether a supernova
occurring out of the ecliptic plane and focused more on the higher latitudes where transport is
more efficient, could have a greater impact. Extrapolating from Reddmann et al., (2023) we
argue that having greater ionization at higher latitudes above 30 km is still inefficient for
destroying global ozone which is concentrated at lower latitudes and at altitudes below 30 km.
By contrast, other phenomena linked to supernovae, such as gamma rays and cosmic rays, are
known to be absorbed by the atmosphere near the peak of the ozone layer in the 20-30 km
altitude range (Melott et al., 2017) and at lower latitudes. Therefore, in our assessment, those are
likelier candidates for causing global ozone destruction that would greatly enhance the flux of
destructive UV radiation to the surface. However, we should conclude by noting that even in
those cases, the destructiveness of both the gamma ray and cosmic ray mechanisms have also
been recently called into question (Christoudias et al., 2024). Our calculations here are therefore
consistent with Christoudias et al., (2024) in showing how the earth's atmosphere can shield its
biosphere.

*Code and Data Availability.* The TIME-GCM code is available by contacting the National Center for
Atmospheric Research. The model output produced herein is reproducible from the TIME-GCM model
source code following the discussions and implementations of the nudging schemes and lower boundary
conditions described thoroughly in Sections 2.4 and in Jones Jr. et al. (2018) and Jones Jr. et al. (2020).
Daily NCAR TGCMs outputs in netCDF format from this study are archived on the DoD HPCMP long-
term storage system. MERRA-2 middle atmospheric horizontal winds and temperatures used for
constraining TIME-GCM dynamics are available at https://disc.gsfc.nasa.gov/datasets?project=MERRA-
2. The SABER and MLS data used in Figure 9 were respectively obtained from https://saber.gats-
inc.com/ and https://mls.jpl.nasa.gov/eos-aura-mls/data.php. Other model output such as CHEM1D and
specific supernova output from TIMEGCM are both available in separately labeled folders on
https://map.nrl.navy.mil/map/pub/nrl/. The /chem1d folder contains the source code of the model and
there are text files for running the supernova simulations. The /timegcm_supernova folder contains
python compatible IDL save files of both TIMEGCM output and the NRLFLARE simulations along with
text files describing them.
*Author Contributions.* DES conceived the study, performed the analysis of the TIMEGCM
output, conducted the CHEM1D analysis and led the writing. MJJr. configured the TIMEGCM, both to be
nudged by MERRA and to input the NRLFLARE spectra, performed the simulations and wrote Section
2.2. JWR is the developer of NRLFLARE; he provided the soft X-ray spectra used by the TIMEGCM and
wrote Section 2.1.
*Competing Interests* The contact author has declared that none of the authors has any competing interests.
*Acknowledgements*. This work was supported by the Office of Naval Research. We also acknowledge the
NASA Living with a Star program for supporting development of NRLFLARE and the development of
the supernovae soft X-ray spectra. Computational resources for this work were provided by the U.S.
Department of Defense (DoD) High Performance Computing Modernization Program (HPCMP).

*Financial support.* This research has been supported by the Office of Naval Research (6.1
funding).

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
