# Peer review of "nitric oxide and stratospheric ozone"

_Annales Geophysicae, 2024_

## Author Comment (AC2)

**Response to Reviewer #2.**

Note, it is new to us to have to send a response without an accompanying revised manuscript. Given that we made some important changes/additions, it seems like it could be confusing to interpret our without the manuscript. We therefore added the new text and figure into the response below. Hopefully with the different fonts it will be not hard to follow what we're saying. Bold face red is our response to each comment. Bold face black "Times New Roman" font is the new text. Unbolded black "Helvetica" is the reviewer's initial comments.
* * *
**Review of the paper by Siskind et al.**

The paper examines the impact of X-ray emission of nearby young supernova remnants (SNR). Using a realistic model of the middle atmosphere and lower thermosphere including photochemistry, the authors test for planet Earth the hypothesis brought up by Brunton et al. who found a possible threat to biospheres in the wider cosmic neighbourhood of such SNRs by the prolonged strong X-ray emission phase. The paper shows that the rough estimation of Brunton et al. for the lethal distance between a SNR and the impacted biosphere does not hold and that in the case of Earth its atmosphere effectively shields also against the threat by extended X-ray emission from young SNRs.

The paper is absolutely in time and is a very valuable contribution to the field of harsh environments for early life and existence of biospheres in stellar system with habitable planets. The methods applied are essentially sound, even when the authors deduce their esimation of the NOx input and related ozone loss from some kind of extrapolation.

I only have a few general comments:

**Response: We thank the reviewer for some very thoughtful comments that we think led to some clear improvements in the paper. Before we reply to each comment specifically, we summarize our overall response. As noted below, we perhaps were not clear enough that our work is really just focused on, and limited to, NOx production by soft X-rays that are absorbed in the mesosphere- well above the ozone layer. As such, we concur with the reviewer that our title was too general and we have changed it to e refer explicitly to soft X-rays. The new title will be:**

**Effects of supernova induced soft X-rays on middle and upper atmospheric nitric oxide and stratospheric ozone**

**And indeed, Brunton mention the limitations underlying this assumption and, importantly, state that this "complicates the extrapolation" from older work that combined both soft and hard spectra (e.g Ejzak et al). We added the Ejzak et al., and Reddmann et al. references offered by the reviewer and have added discussions about what other scenarios might do- for example, harder spectra or supernovae outside of the ecliptic plane.**

**A second major change is that, following the reviewer's third suggestion below, we have added a figure (Figure 4) which shows the production of NOx in terms of gigamols (GM), both for the stratosphere and mesosphere. This then allowed us to make some more quantitative comparisons with other space weather estimates (Vitt and Jackman; Reddmann) and we thank the reviewer for that suggestion. Below we attach the new figure and the extended discussion of that figure which now goes at the beginning of Section 3.**

[Figure]

**Figure 4. Total globally integrated NOx (=NO + NO2) number of molecules (GM: gigamoles) for the baseline no flare case (dashed line) and the continuous soft X-ray flare (solid line) for the mesosphere (top panel) and stratosphere (bottom). The soft X-ray event, which assumes a flare spectrum from the Sept 10, 2017 flare is assumed to have begun on that day (day 253 of 2017).**

**3. Seasonal Variation of the Xray-IE in the middle atmosphere**

In order to provide a broad, but quantitative, overview of the production of NOx from the extended flare/supernova, Figure 4 shows the calculated total number of NOx molecules in units of gigamoles (GM) and compares it to a baseline/no flare simulation. This quantity

has been previously used (Vitt and Jackman, 1996; Siskind et al., 2000; Funke et al., 2005) as a way of quantifying space weather impacts on the ambient NOx budget. Here, the production of NOx is mostly in the mesosphere while the impacts on ozone are in the stratosphere. Therefore, using the 50 km level as an arbitrary dividing line, we break out our calculation to illustrate mesospheric NOx (top panel of Figure 4) and stratospheric NOx (bottom panel of Figure 4) separately.

In each panel, the upper (solid) curve is the NOx with the extended flare calculation. The dashed curve is a baseline case with no flare. First, considering the no flare case, our stratospheric value equilibrates to around 20-22 GM (we attribute the initial decrease to an excess of NOx in the initial conditions). Given that the model bottom boundary is 30 km and that significant NOx lies below 30 km, our result is likely consistent with previous estimates by Vitt and Jackman (1996) of 29-30 GM for the stratospheric production of NOx from $N_2O$ oxidation. For the no flare case, the upper panel shows a value between 3-5.5 GM due to the background secondary NOx maximum in the upper mesosphere/lower thermosphere.

For the flare case, the mesospheric results show a rapid increase to over 15 GMs. The stratospheric NOx does not increase immediately, but as evidenced by the increasing divergence between solid and dashed curves, shows a gradual increase in the flare produced NOx. It is interesting that for all 4 curves, the maximum NOx occurs in the period from days 570-620. This corresponds to August and September and coincides with the late winter period in the Southern Hemisphere. As we will discuss, satellite analyses have indicated that the maximum delivery of upper mesospheric/lower thermospheric NOx to the stratosphere occurs during that time and, as we show below, this is indeed the case here.

Finally, we can give a crude comparison of the global effects of this extended flare to previous space weather phenomena. The largest difference in the stratosphere between the flare and baseline, as shown in the bottom panel of Figure 4, is ~4.5 GM. This can be compared to the 1.3 GM that Funke et al., (2005) estimated was delivered to the upper stratosphere during the 2003 Antarctic winter which followed a period of elevated space weather activity. Thus the extended flare appears to exceed that by about a factor of 3.5. Funke et al., (2005) also estimated a roughly equivalent amount of NOx would end up in the lower stratospheric polar vortex, below our 30 km bottom boundary. Siskind et al., (2000) also estimated a peak vortex amount of about 0.8-1.3 GM. If we assume this rough equivalence between upper stratospheric and lower stratospheric polar vortex delivery applies here, then we arrive at an estimate of 9 GM from this extended X13 flare. By comparison, Vitt and Jackman (1986) estimated a total production of 7 GM from the large solar proton event in 1989. Thus our current simulation exceeds any previously documented space weather effect on stratospheric NOx, but at the same time, it is not dramatically bigger. As we shall see when we look at the details of the NOx distribution and its effects on ozone, our results follow that pattern i.e., greater, but not dramatically so.

Back to reviewers comments
* * *
1. The authors use the TIME-GCM for their study. The authors state themselves that the model in this configuration is not able to simulate elevated stratosphere events, which are key for strong NOx intrusions in the NH, and probably underestimates downward transport inside the polar winter vortex in the lower mesosphere in general. The authors comment that comparisons with MIPAS data show good agreement for midlatitudes instead, but these airmasses see the Sun also during winter and NOx here has a limited lifetime. Somehow related, NOx transported below the lower boundary is lost in their model. But during summer with the change of the circulation this NOx is brought to the middle stratosphere again where it could contribute to ozone loss. Can the authors give an estimation of this contribution which is lost in the model?

**Response: It is hard for us to directly quantify the effects of the bottom boundary in the way that the reviewer is seeking. There is no discussion in the literature of which we am aware of enhanced NO from space weather (for example SPEs) being created below 30 km and then getting lofted upwards. Generally, considering the Brewer Dobson circulation- polar NO that is not mixed equatorward in the 40 km region will mix into the troposphere.**

In addition, the second, strong case does not run through and the authors must rely on some reasonable extrapolation. Despite the authors state this clearly, perhaps other models like WACCM are obviously better suited to study such events if the shortwave photolysis part developed by Siskind would have been implemented there.

2. Despite the rather large energy input, the modelled impact on the ozone layer is small. Typical particle events connected to magnetic storms show hemispheric power values of a few hundred GigaWatts for several hours which corresponds to energy input around 50 J/m2. Obviously, the energy spectrum of the ionizing radiation or particles is most important for the impact in the stratosphere. Brunton et al. speculate (section 3.1) that the spectrum of young SNRs may be significantly harder than assumed in this paper. This would result in deeper penetration into the middle atmosphere and possibly could strongly enhance longlived NOx in the stratosphere as then NOx could bridge the gap of photolysis loss descending NOx faces in spring. The authors should point to that critical uncertainty. Of course, observations in the extended spectral range are needed to further reduce this uncertainty, but also some sensitivity model study would be helpful. In this context, a

3. The authors should estimate the total amount of NOx brought into the atmosphere, not just for the MLT. For comparison, particle events like the Halloween storm show inputs of about 2 Gmol of NOx, and a recent study by Reddmann et al. https://doi.org/10.35097/1104, modelling an extreme SPE/storm event with an input of 30 Gmol also show limited impact on the ozone layer, in line with this study.

**Response:  As noted above, we have responded to this comment with the new Figure 4. We also added some discussion in the Conclusions section. As noted by the Reviewer, 30 GM, even when focused more on the high latitudes (i.e., out of the**

**ecliptic plane) where descent might be more efficient and shielded from sunlight, still doesn't cause biospherically significant ozone destruction.**

**New text in the Conclusion**

**. More recently, Reddman et al., (2023) performed a similar simulation of an extreme solar proton event combined with an extreme geomagnetic storm. They show dramatically enhanced ionization in the high latitude regions for all altitudes above 30 km. Their extrapolated NOx production is on the order of 25-30 GM roughly equivalent to our extrapolation for our X27 case, but now occurring directly at higher latitudes where transport to the lower stratosphere might be hypothesized as more efficient. However, like our results, they find the overall impact of any resulting ozone reduction on UV flux to the surface to be limited to less than 5%. The Reddman simulation is important because it might be relevant to the question of whether a supernova occurring out of the ecliptic plane and focused more on the higher latitudes where transport is more efficient, could have a greater impact. Extrapolating from Reddmann et al., (2023) we argue that having greater ionization at higher latitudes above 30 km is still inefficient for destroying global ozone which is concentrated at lower latitudes and at altitudes below 30 km.**

4. The position of the source in ecliptic plane may be not the position with maximum impact. Could the authors try to have a source at the celestial poles? This would put the NOx enhancement deeper into the atmosphere due to the vertical infall and could bring Nox to also the stratosphere.

**Response: While it would a major effort for us to re-run the model, we argue that our discussion of the Reddmann results referred to above addresses this. It still doesn't seem to be a critical factor.**

5. The results of this paper is at strong odds with the result of Brunton et al.. For further studies it would be helpful, if the authors could explain why the results of their study and that of Brunton et al. differ so much. A possible reason for the differences could be the hard X-ray spectrum part. Can you compare your spectrum with Fig. 1 of Ejzak et al?

**Response: As noted above and shown below, we now discuss these differences in Section 2.1. We should add that Brunton et al. were not doing a self-consistent atmospheric calculation. They were just extrapolating from older results and as they note, Ejzak did include higher energy photons in their spectrum- which we do not (and hopefully our paper title is no longer misleading)**

**New text in Section 2.1**

**A key assumption is that we are essentially ignoring wavelengths less than 0.05 nm. As discussed by Brunton et al. (2023) these wavelengths would be absorbed much more directly into the stratospheric ozone layer. Older studies (cf. Ejzak et al., 2007) did include**

these wavelengths and this inclusion, as noted by Brunton et al. "complicates any direct extrapolation" of those results when considering a purely soft X-ray event, as we do here. Our work is the first to use a model of the stratosphere, mesosphere and thermosphere to explicitly consider how the indirect effects of enhanced soft X-rays could affect global ozone.

6. The labeling of most figures does not confirm with standard. Often the y-axis shows the unit but not the quantity or vs versa; how to show units (with brackets or without) is not consistent.

**Response:  Sorry for the confusion- we were admittedly a bit sloppy. For the revision, we added a more complete y axis label for Figure 8 so it's clear it's the same quantity as Figure 9. We also added more complete labels for Figure 10 (ozone profiles with SABER)**

Minor comments and typos:

Title: "Supernova effects" is rather unspecific. Perhaps "No threat to the ozone layer by X-ray luminous SNs" ?

**We have changed the title as we discussed above.**

L14 "planetary": The paper only deals with the Earth, so perhaps "ozone layers of Earth-like planets"

**done**

L24 "most global": "strongest global"?

**done**

L35 "these": delete

**done**

L88 "Thus our": delete "our"

**done**

L117 "For our purposes ...": This is too general

**Changed- see lines 117-118, new text reads:**

**For the purposes of calculating NO production, the exact spectral shape is less important than the total soft X-ray energy input driving the atmospheric response.**

L156: "our spectrum" replace "our"

**———Changed to "NRLFLARE"**

L158ff: "well covered with modern spectra" "suggests that .... agrees" : please reformulate to be more concise

**Done (hopefully). New text:**

**Comparing our results in detail with Rodgers et al. (2010) yields good agreement with our calculated 0.1 – 1 nm flux of .004 W/m$^2$.**

As discussed by Siskind et al., (2022) this seems consistent with Orbiting Solar Observatory (OSO) data presented by Neupert et al., (1967), although this spectral region is not well covered with modern spectra

L232: Kp = 3 (L235) still shows some particle ionization. **The idea is that it's constant**.

L236: 30°

**We don't think this this comment is correct. Cf.
https://astronomy.swin.edu.au/cosmos/G/Galactic+Plane**

L276: Michelson

**Corrected**

L289: "descent" is purely dynamic. You mean the amount which descends.

Added "**of NOx**"

L348, Fig. 5: Why not showing diffs?

**We made plots of the differences but felt that showing the absolute abundances gives better context, esp. given all the discussion about MIPAS data. And the discussion on lines 400-401, in our opinion, clarify exactly what those differences are. No change was made here.**

L361, "1.0": unit?

**Sorry, the text might have been a bit confusing. As stated in the caption, it's a ratio. Text has now been changed (line 419) and now reads:**

**The ratios are less than 1.0 globally for the entire year, which means lower ozone for the X13 simulation.**